# A Systematic Review of Mechanisms Underpinning Psychological Change Following Nature Exposure in an Adolescent Population

**DOI:** 10.3390/ijerph191912649

**Published:** 2022-10-03

**Authors:** Megan Rowley, Raluca Topciu, Matthew Owens

**Affiliations:** 1CEDAR (Clinical Education Development and Research), University of Exeter, Exeter EX4 4QQ, UK; 2The ROWAN Group, Exeter EX4 4QQ, UK; 3Somerset Foundation Trust NHS, Taunton TA1 5DA, UK; 4Department of Psychology, University of Exeter, Exeter EX4 4QQ, UK

**Keywords:** nature exposure, mechanisms, adolescence, systematic review, mental health, prevention

## Abstract

This systematic review aimed to identify mechanisms of psychological change following exposure to nature within an adolescent population. Keyword searches within Scopus, PsychINFO and Web of Science were carried out to include articles published by 14 September 2021. Records were reviewed in line with inclusion criteria: samples with an average age of 24 and under, exposure to nature vs. control using an experimental or quasi-experimental design and outcomes of mental health and psychological status. The review resulted in 27 papers that were assessed for methodological quality and manually searched for mediation analyses. A range of psychological outcomes were identified and grouped into 10 categories: Mood and Affect, Mental Health, Wellbeing, Perceived Restoration, Stress, Energy, Cognitive Functioning, Resilience, Self-Concept and Pro-Social Behaviour. Only one formal mediation analysis was reported, highlighting a mediating role of belonging in increases in resilience. Limitations include the majority use of university student samples and over half of the papers being of low methodological quality. No firm conclusions on key mechanisms in an adolescent population were made due to insufficient evidence of mediating variables. The development of methodologically rigorous experimental studies with the inclusion of statistical pathway modelling is needed to test and specify plausible mechanisms.

## 1. Introduction

Adolescence presents a key developmental window for the prevention of mental health disorders, highlighting it as a critical time to intervene [1]. Exposure to nature is a modifiable lifestyle behaviour that shows promise in supporting the prevention of mental health difficulties given its benefits to health and wellbeing [2,3]. A central aspect of health research is to identify mechanisms that account for the observed benefits, which is invaluable to the design and testing of strategies to promote the beneficial effects of nature exposure [4]. As such, there is a need to understand why nature exposure leads to psychological benefits in an adolescent population. Specifying and investigating these key underlying mechanisms of change will be imperative in developing effective nature-based interventions for mental health and wellbeing in this period of development. Furthermore, it is particularly important to identify any key mechanisms that may also be implicated in the development and maintenance of mental health difficulties to enable the best chance of mitigating against the risk factors for the development of disorders [5]. Whilst previous reviews have aimed to explore psychological outcomes following nature contact in children and adolescents [6,7], further investigation is needed specifically in an adolescent population, with a consideration of suggested mechanisms of change. 

Adolescence presents a period of significant vulnerability to mental health problems with 75% of adult mental health disorders emerging before 25 years of age [8,9,10]. The experience of mental ill health in adolescence, such as depression, has significant consequences on health in young adulthood and places increased burden on society through associated direct and indirect costs [11]. As such, adolescence highlights a key developmental window for the intervention of mental health difficulties, aimed at averting their course [1,12]. 

Given the heightened potential for prevention in adolescence as a period of development, targeting lifestyle behaviours as modifiable factors associated with common mental health disorders has been suggested as a key intervention strategy [13]. Lifestyle psychiatry is an approach that aims to facilitate the management of mental health disorders via a holistic approach to health [14]. Whilst this has traditionally focused on areas such as physical activity, diet and nutrition, adequate sleep, and stress management, it also offers the opportunity for immersion in nature. With a growing evidence base for the impact of nature on health and wellbeing [2,3], and a sound evidence base for the emotional benefits of exposure to nature [15], there has been a call to focus on nature as a key, targetable lifestyle factor in clinical practice [16]. 

Nature can be defined as “areas containing elements of living systems that include plants and nonhuman animals across a range of scales and degrees of human management” [17]. The connection between nature and health and mental wellbeing is underpinned by a suggested affinity to nature as outlined in E.O. Wilson’s Biophilia Hypothesis [18]. Biophilia, literally meaning a ‘love of life’, states that humans have a connection to nature that is both adaptive in response to evolving in natural environments and innate in origin [19].

Nature has been found to directly promote multiple aspects of health, including mental health and wellbeing [2,3]. The psychological benefits associated with exposure to nature have been particularly highlighted and encompass emotional and cognitive functioning, wellbeing and other dimensions of mental health, such as depressive symptoms, anxiety, affect and stress [3,15]. More specifically, in children and adolescents, nature exposure has been found to lead to improvements in mental health outcomes including mood, emotional wellbeing, resilience and stress, alongside beneficial improvements in attention and memory [20,21,22,23,24,25]. 

Vanaken and Danckaerts [21], for example, found credible support for a confounder-adjusted (demographic and socio-economic) relationship between green space exposure and general mental health, as indicated by research using the Strengths and Difficulties Questionnaire, supporting the association between nature and mental health independent of confounders that may also predict mental health. In another review, Mygind et al. [25] found that, when focusing specifically on studies that compared immersive nature experiences with a control group, outcomes such as self-esteem and resilience were enhanced compared to comparison conditions. However, they concluded that these findings only conditionally provide recommendations for practice due to the low quality of evidence. With possible benefits highlighted, developing nature-based interventions aimed at supporting wellbeing and other dimensions of mental health has the potential to prevent the burden associated with mental ill health that develops in adolescence. Whilst the specific psychological mechanisms are less than clear, there has been a number of suggested theories through which nature has positive effects via psychological and psychophysiological recovery.

### 1.1. Stress Reduction Theory (SRT; [26])

SRT suggests that nature promotes psychophysiological restoration through a reduction in stress and negative affect, alongside a shift to a more positively-toned emotional state. Believed to be grounded in the psycho-evolutionary theory of biophilia, exposure to unthreatening natural environments is posited to lead to stress-reducing psychophysiological responses, as a result of positive and pre-cognitive responses to nature [15,27]. More specifically, unthreatening natural settings are believed to trigger an initial positive affective response. This, combined with the mobilisation of the parasympathetic nervous system, leads to higher levels of positive feelings, reduced negatively-toned feelings and a maintenance or restoration of attention and energy, which are adaptive to survival [28,29]. Evidence for SRT following nature exposure is supported by the positive effects on various physiological and emotional parameters associated with stress recovery [30,31]. These outcomes have been particularly highlighted in the forest-bathing (or ‘shinrin-yoku’ in Japanese) literature [32]. 

### 1.2. Attention Restoration Theory (ART; [33])

ART suggests that directed attention resources are replenished when in nature due to our involuntary attention resources engaging with intrinsically fascinating stimuli in the environment. More specifically, nature can be considered a restorative environment as it enables: (1) a sense of being away and thus a shift away from mental activity that requires directed attention, (2) soft fascination which is attentionally effortless, (3) a sense of extent, which enables the mind to be engaged with seeing and experiencing nature and (4) a high compatibility between the environment and an individual’s purpose [34]. Posited to lead to cognitive clarity and reduced mental fatigue [35], research also supports restoration in cognitive domains, such as cognitive flexibility, working memory and attentional control, enabling greater effectiveness [36]. 

### 1.3. Affect Regulation

More recently, an evolutionary model of affect regulation and neurophysiology of emotion has been posited, in which nature-based wellbeing benefits are experienced through an emotional state balance [27,37]. The model outlines how nature exposure brings emotional balance through activating certain aspects of our nervous system [38]. Furthermore, nature plays a role in increasing adaptive emotional regulation strategies and decreasing maladaptive ones, depending on the characteristics of and cues in the environment [39]. Support for a model of affect regulation comes from evidence of individuals seeking natural environments to support their emotional regulation [40,41] and rumination reduction following nature contact [42], with evidence of this mediating the relationship between nature and negative affect [27,43,44,45]. 

As over-arching theories, SRT, ART and affect regulation provide candidates for key mechanisms of psychological change following nature contact. Research also suggests that there are additional constructs that may interact with or be independent of these theories. Firstly, perceived restorativeness [46,47,48] has been suggested, which is a measure of the restorative components of natural environments as indicated by ART, but is not necessarily predictive of or necessary for attention restoration [49]. Secondly, nature connectedness as a psycho-evolutionary need to affiliate with the natural environment has also been suggested as an additional construct [50]. More recent and emerging theories suggest that there are other potential mechanisms of change between exposure to nature and psychological outcomes, some of which are of particular interest as they are also implicated in common mental health difficulties. These include: sleep [51], mindfulness [42], with support for it mediating the association between nature connectedness and wellbeing [52,53,54,55], and physical activity [56,57]. Other key factors relating to psychological change and wellbeing have also been implicated, such as self-esteem, with mixed outcomes, but possible effects via physical activity [58,59] and social cohesion [60,61]. 

Some mechanisms, in particular those that may also be implicated in the development and maintenance of mental health, such as mood and affect, stress, rumination, sleep, etc. [62,63,64,65] could be important, clinically relevant, modifiable mechanisms. Exploring these in the adolescent literature is imperative to the ongoing development of nature-based interventions aimed at supporting mental health and wellbeing specifically in this population.

Finally, whilst there is evidence for the beneficial effects of nature in adolescents, it is important to acknowledge that certain types of nature may not be adaptive for everyone. In recognising the relationship between human beings and the environment as favourable for survival, it is important to acknowledge the negative evolutionary feature, biophobia [66]. Biophobia can be considered as a fear of natural elements, allowing for a safe reaction to potential threats in the environment [67]. Some natural environments or features, therefore, may be aversive to some. Furthermore, childhood nature experiences and direct experiences which evoked negative perceptions of nature have been found to predict disgust sensitivity to and fear expectancy of nature in young adulthood, meaning that some adolescents may have reduced preferences for nature [68]. 

### 1.4. Aim 

The aim of the current review was to identify any mechanisms of psychological change following exposure to nature within an adolescent population. 

## 2. Methods and Analysis

This systematic review is in accordance with the Preferred Reporting Items for Systematic Reviews and Meta-Analyses (PRISMA) guidelines [69]. 

### 2.1. Search Strategy 

Comprehensive electronic searches were conducted up to 14 September 2021 in three major databases: Scopus, PsychINFO and Web of Science (WOS). The search strategy used Boolean operators to combine key words indicative of nature, mental health and adolescents (Table 1). Individual terms were searched in the titles, abstract, and/or keywords of research articles. Search strategies for each database can be found in the supplementary material (File S1). Research papers were manually reviewed for the inclusion of mediation analyses. Reference lists of articles screened at the full text stage were manually checked to find any additional publications relevant to the search question. 

### 2.2. Eligibility and Study Selection

Following the removal of duplicate studies, all records were reviewed based on the title and abstract, in line with the inclusion and exclusion criteria outlined in Table 2 using the PICO tool (population, intervention, comparison, outcomes and study design [70]). The population age parameter was set according to Sawyer and colleagues [71] definition of adolescence ranging from age 10–24 years. Studies with participants over 24 years were included as long as the average age of the sample was 24 or younger. Research that included a therapeutic intervention, such as wilderness-based therapies, outdoor psychotherapy and outdoor behavioural healthcare, were not included in order to distinguish the impact of nature from the impact of therapeutic intervention on mental health. Beyond the PICOs question, research studies were limited to those published since 1990, in peer-reviewed journals and in the English language. All studies identified as potentially relevant were then reviewed in full text. A second independent reviewer reviewed six studies (20%) at the full text stage based on inclusion and exclusion criteria which resulted in full agreement. 

### 2.3. Data Extraction 

The following data were extracted using a pre-designed data extraction form: author(s), year of publication, country of study, population, sample size, mean age of sample, gender of participants, study design, type of nature exposure (experimental condition), comparison condition, outcome measures, data collection time points, main effects and inclusion of mediation analysis. Study authors were contacted where necessary information was missing. 

### 2.4. Study Quality and Risk of Bias

Research articles were assessed using the Effective Public Health Practice Project (EPHPP) quality assessment tool [72]. The EPHPP tool assesses methodological quality on the basis of six components: selection bias, study design, confounders, blinding, data collection methods and withdrawals and drop-outs. The component sections are rated according to three outcomes: strong, moderate and weak. The overall global rating for each article is then classified according to the number of “weak” ratings it received, whereby strong is considered to have no weak ratings, moderate has one weak rating and weak has two or more weak ratings. A second independent reviewer reviewed the methodological quality of three papers. A weighted comparison of the reviewer ratings suggested substantial inter-rater agreement (Kappa = 0.71, *p* < 0.001). Any discrepancies were resolved through a discussion and this subsequently led to the overall quality rating of one paper to be amended. The quality assessment tool was used in line with the authors’ recommendations whereby outcomes of only strong and moderately rated studies were included in the review [72]. 

### 2.5. Data Synthesis

Main results from strong and moderately rated studies were extracted and organised into categories of outcomes on the basis of outcome measures used and the measurement of similar concepts. This iterative process involved discussion of suitable categories within the research team. Outcomes were then grouped into significant and non-significant findings within each category of methodological quality. A meta-analysis was not conducted due to large heterogeneity in outcomes and study characteristics including types of natural environments, types of nature contact and gender of participants. 

## 3. Results

### 3.1. Search Results

A flowchart indicating the search outcome at each stage of the process can be seen in Figure 1. A total of 76 papers were included for full-text review and 37 met eligibility criteria; however, a further 10 studies were excluded on the basis of using a semantic differential method [73] as the sole psychological outcome measure, as these articles showed insufficient evidence of the development and validity of their primary outcomes, leading to uncertainty of the measurement rigour. This resulted in the inclusion of 27 papers for review.

### 3.2. Study Characteristics 

The characteristics of the 27 included papers in line with the PICOs criteria are shown in Table 3. All studies were published between 1996 and 2021 and were carried out across three continents: Asia, Australasia and Europe. When considering study designs, 20 studies adopted an experimental design and seven used a quasi-experimental design, whereby the method of allocating participants to conditions was not under the control of the investigators and therefore was not randomised. For comparison groups, 20 of the designs had an active control condition, six were education as usual and one wait-list for intervention. One study did not include a non-nature control. When considering study population characteristics, the mean age of samples were all under 24 years of age. Participants in eight studies were school aged students (13–18) and the remaining studies recruited young adults age 18+. Overall sample sizes ranged from 13 to 396, 59% of them with under 70 participants. One study used a female only sample, 11 a male only sample and 15 recruited mixed genders. The type of nature participants were exposed to varied greatly, as did the type of contact with nature. Nineteen studies used a one-off exposure to nature, six used engagement in activities in nature and one used repeated access to nature. All studies collected data pre and post nature exposure, with three studies collecting data at a follow up point between 5–12 months post nature exposure. 

The examined studies identified 60 significant outcomes of psychological change following exposure to nature (Table 4). All outcomes were assigned to one of 10 categories: Mood and Affect, Mental Health, Wellbeing, Perceived Restoration, Stress, Energy, Cognitive Functioning, Resilience, Self-concept and Pro-social Behaviour. A summary of the measurement tools used in each study and a summary of papers by outcomes can be found in the Appendix A.

### 3.3. Quality Appraisal 

Quality ratings for the studies are indicated in Table 3. Eleven studies were rated of a moderate quality and 16 studies were rated as weak quality. Component ratings within each individual study can be found in the supplementary material (Appendix A). Fifteen out of 17 weakly rated studies were rated as weak for the blinding component rating. 

### 3.4. Outcomes

#### 3.4.1. Mood and Affect 

After the removal of studies rated of a weak quality, negative mood states were found to significantly reduce in four studies (#1, #5, #6, #16) and positive mood states were found to significantly increase in three studies (#5, #6, #16). Where effect sizes were reported, a large effect was indicated. In these studies, participants were all university students exposed to forest environments. The POMs was used as the main outcome measure for mood, which aims to assess a variety of mood states: Tension, Depression, Anger, Vigour, Fatigue and Confusion. Takayama et al. [89] found significant changes on all subscales except anger and depression following forest walking and viewing, however, the authors used a shortened version of the POMs (POMs; [101]). Positive affect was found to be significantly higher following nature contact in five studies (#1, #5, #6, #7, #19), with a medium effect as reported in one study only. Hartig et al. [80] found that, in a natural environment, participants who did not complete a task vs. participants completing a task requiring attention focus showed greater increases in positive affect, suggesting that the task worked against positive emotions. Negative affect was significantly lower in one study of an all-male sample which found a large effect (#16); however, three studies (#1, #5, #9), including Bielinis et al. [74] who recruited female participants only, did not find any significant differences. 

#### 3.4.2. Stress

Four studies found significant decreases in stress using a variety of outcome measures after the removal of weak studies (#17, #21, #23, #7). Tsunetsugu et al. [90] and Wang et al. [93] found heart rate to be slower after exposure to nature compared to urban environments, with medium effect sizes reported. Kelz et al. [96] and Tsunetsugu et al. [90] found measures of blood pressure to decrease following nature exposure, with small to medium effects; however, in the latter study, only significant differences in diastolic blood pressure were seen. Wang et al. [94] also measured skin conductance which was found to reduce after viewing nature-based scenes, with a large effect reported. Hartig et al. [80] found viewing nature after completing a drive or task led to a more rapid decrease in diastolic blood pressure. Furthermore, walking in nature vs. an urban setting initially led to a change in blood pressure suggestive of stress reduction, but this effect dissipated over time. 

#### 3.4.3. Energy 

After removing methodologically weak studies, four papers found significant increases in measures of vitality (#1, #5, #6, #16) with a medium effect as reported in one study only. One paper reported significantly more self-reported feelings of being refreshed following exposure to forest environments, with a large effect (#17). An additional study by Fuegen and Breitenbecher [92] found that exercise or rest sessions outside led to increased energy and decreases in tiredness compared to exercise or rest indoors sessions, with a small-medium effect. Indoor sessions, however, also included exposure to the outdoor scene via a visual stimulus; therefore, changes in energy and tiredness were not compared to a non-nature control, limiting the conclusions that can be made.

#### 3.4.4. Perceived Restoration 

After removing one study of weak quality, six papers found significant increases in perceived restoration (#1, #5, #6, #16, #21, #23). Where effect sizes were reported, a medium to large effect was indicated. Four of these studies used the Restorative Outcome Scale (ROS) (#1, #5, #6, #16), whereas two used the Perceived Restorativeness Scale (PRS) (#21, #23). Beilinis et al. [79] found that even in winter, when no leaves were present on the trees, 15 min of forest-bathing compared to viewing an urban environment led to increases in self-reported restoration. 

#### 3.4.5. Mental Health 

After the removal of methodologically weak studies, only one study reported a change in mental health using a measure of anxiety. Specifically, following a stress induction procedure in a Chinese sample involving a mock spoken-English exam, Wang et al. [93] found a significant decline in state anxiety after viewing nature-based urban park scenes, with a large effect reported. 

#### 3.4.6. Self-Concept 

Self-concept was defined by measures of self-esteem and self-efficacy. After the removal of weak rated studies, no studies reported significant increases in self-concept. Wood et al. [58] specifically looked at self-esteem following exercise whilst viewing natural or built scenes. Whilst they found self-esteem to be improved through physical activity alone, there was no effect of viewing natural scenes. 

#### 3.4.7. Cognitive Functioning

Following the removal of weak studies, two studies focusing on changes in attention indicated significant findings (#7, #21). Where effect sizes were reported, a medium to large effect was indicated. However, in one of these studies [80], increases in attention appeared to be due to performance decrements in the urban environment as opposed to performance increases in the nature environment. Two studies reported non-significant findings (#19, #23). 

#### 3.4.8. Wellbeing

Only one study exploring wellbeing was rated greater than weak quality. Specifically, in their quasi-experimental study, Kelz et al. [96] found psychological wellbeing to increase on two different outcome measures after exposure to a nature-based schoolyard designed to include more greenery, seating and sporting equipment. Pre and post measures were taken over a seven-week period and compared to pupils at two schools without schoolyard changes. Small effect sizes were indicated. 

#### 3.4.9. Resilience 

After removing studies of weak quality, one quasi-experimental study, as further described in the mediation section below, reported increases in resilience in high school participants following a 10-day voyage on a ship which was maintained at a 9 month follow up (#25). A large effect was reported. 

#### 3.4.10. Pro-Social Behaviour 

Only one study explored the impact of nature exposure on pro-social behaviour between those who completed an outdoor education programme and controls. However, this was rated weak quality and no significant findings were found (#18). 

### 3.5. Mediation Analyses 

To consider any key indicated mechanisms of change, papers were manually searched for the inclusion of mediation analyses. Only one study used a formal analysis. Specifically, Scarf et al. [98] carried out multiple and serial mediation analyses and found that sense of belonging directly contributed to increases in resilience following a 10-day sailing voyage and centrality of identity indirectly predicted resilience immediately post voyage and at a 9 month follow up via belonging. 

Hartig et al. [80] used correlation analyses and found changes in attentional performance to correlate with changes in positive affect after partialling out the effects on environment. However, in absence of a formal mediation analysis, it is not possible to make any conclusions about casual mediators [102].

### 3.6. Risk of Bias

Only 11 studies out of the identified 27 were deemed to be of a moderate methodological quality and, even in these studies, important risks of bias were noted. Nine studies recruited participants who were only somewhat likely to be representative of an adolescent population. Eight of these studies used university students or young adult populations (#1, #5, #6, #7, #16, #17, #19, #21) and Scarf et al. [98] did not make it clear what percentage of selected individuals agreed to participate in the study. Two studies were quasi-experimental in their design and therefore did not randomise participants (#24, #26). Five studies controlled for less than 80% of confounders or did not report whether relevant confounders were controlled for (#1, #7, #17, #21, #25). In eight of the studies the blinding procedure was not described or participants and assessors were not blind to the intervention (#7, #16, #17, #19, #21, #22, #23, #25). 

## 4. Discussion

The aim of the current review was to identify mechanisms of psychological change in adolescents following exposure to nature. Through a systematic review of the literature, 27 papers were selected. All studies had a mean sample age of equal or less than 24 years and included control conditions. Methodological quality reviewing identified 11 papers of a moderate quality. Outcomes associated with psychological change were then identified and subsequently assigned to 11 categories. 

Mood and affect was the most widely studied outcome of psychological change, with more significant than non-significant findings, except for measures of negative affect which demonstrated more non-significant findings. Both Zhang et al. [23] and Roberts et al. [6] identified improved wellbeing through increases in mood and positive affect following exposure to greenspace and nature-based activities in children and adolescents; however, unlike the current review, they did not identify any papers that specifically measured negative affect. In a meta-analysis of affect following nature contact, in which the majority of the participants were in late adolescence, McMahan and Estes [103] also found nature to improve mental wellbeing through increased positive affect. Similarly, they reported that to a lesser degree nature decreases negative affect, but that the effect was moderated by location of study, type of affect measure and type of nature exposure. It is possible that these study and design related characteristics contributed to the identified heterogeneity of outcomes on measures of negative affect following nature exposure in the current review. 

Just under half of the papers reported outcomes relating to stress and demonstrated more significant findings than non-significant. Studies reporting outcomes on stress showed great variation in the physiological indices used and had a high proportion of weakly rated studies. Whilst improvements in stress have been reported in previous children and adolescents reviews, measures of stress have varied from self-report to objective physiological measurements. Outcomes of stress discussed in the current review captured physiological outcomes only. In a review of adults that distinguished the physiological impact of nature on stress recovery, Corazon et al. [30] also found considerable heterogeneity in outcomes, largely attributed to inappropriate use of physiological measures that were insufficiently sensitive to detect differences. Included in the current review, Wang et al. [94] specifically used a stress-inducing task prior to nature exposure, enabling them to detect differences in stress recovery across different virtual scenes. Stress induction, in a reliable and valid manner, is paramount to exploring psychophysiological mechanisms in the clinical assessment of stress reduction [104]. However, the task used by Wang et al. [94] was not validated and no manipulation check was carried out. Future work exploring stress reduction as a key mechanism of nature exposure should utilise validated stress induction tasks, such as the Trier Social Stress Test [105]. 

All outcomes relating to energy and perceived restoration were significant. Whilst perceived restorativeness has been identified as a mediating variable on psychological outcomes in the adult literature [106], studies included in the current review used two differing measures of restoration. The ROS focuses on self-perceived changes in states and the PRS on the restorative qualities perceived in the environment. Both measurement outcomes, therefore, should not be interpreted together when drawing conclusions on attention restoration and potential mediation effects [47]. Furthermore, a review by Browning et al. [107] found that experimental studies using perceived restoration reported more positive findings than expected, indicating publication bias in reporting the beneficial impacts of nature. Caution should be taken when interpreting perceived restorativeness as a possible mechanism on the basis of differing constructs and possible overreporting. 

Generally, exposure to nature appeared to have a positive impact on outcomes of psychological change in adolescents, with significant findings identified within all categories except for self-concept and prosocial behaviour. Other systematic reviews on the benefits of access to nature on mental health and wellbeing in children and adolescents report similar positive outcomes [6,20,23,24]. However, only Tillmann et al. [7] similarly looked at range of interactions with nature, compared to the other reviews that focused specifically on access to greenspace or nature-based activities only. A lack of significant outcomes for pro-social behaviour following nature exposure in the current review could be considered in light of a previous review by Putra et al. [108], who only found limited evidence for the relationship between pro-social behaviour and greenspace due to it being moderated by socio-demographic factors and gender. Specifically, the association was stronger in children who only spoke English at home, those living in more affluent and remote areas and boys. Similarly, in considering self-concept in the current review, Roberts et al. [6] and Tillmann et al. [7] found inconsistent results for outcomes of self-esteem. This further highlights an inconclusive link between nature exposure and self-esteem in an adolescent population. 

Papers were manually reviewed for the inclusion of mediation analyses to consider any key mechanisms of change. Research using mediation analyses is an important first step for establishing the mechanisms through which an experimental manipulation or intervention leads to change, better ensuring that findings can be generalised into practice through knowing what is needed to optimise outcomes [109]. Methodologically, mediation goes above and beyond demonstrating the overall relationship between two variables by exploring how an additional “third” variable can be incorporated into statistical analyses to undercover why or how the two variables are related [110]. The identification of mediators and mechanisms, particularly in controlled experimental studies of nature exposure, are imperative in clarifying the connection between exposure and the diverse outcomes seen [22,109]. 

In the current review, only one study included a formal mediation analysis. Scarf et al. [97] specifically found a significant direct contribution of belonging on outcomes of resilience and an indirect contribution of centrality of identity, via belonging. This finding speaks to the role of social processes in outcomes of psychological change following nature contact and the study was one of six papers included in the review that used an experimental manipulation of nature exposure carried out with others. Previous research has supported, at least, the partially mediating role of various social processes between nature and psychological outcomes in adults and adolescents, including social interaction and social cohesion, highlighting the salience of social factors [21,111,112,113]. Given that exposure to nature involving social contact creates a complex social environment, there are likely to be a range of potential mechanisms of change and future research should aim to continue to understand how social factors may influence nature contact. Further experimental manipulations of exposure to nature, with and without social contact, will be important to help distinguish the impact of nature experience from social predictors of psychological outcomes. 

In considering other identified mechanisms of change, Hartig et al. [80] alluded to their finding of weak correlations between attention and physiological measures supporting separate processes in line with ART and SRT. Furthermore, a positive correlation between attention and positive affect was found but methodological limitations meant that the authors were unable to carry out a mediation analysis to conclude whether one variable mediates the other. Whilst the conclusions that can be drawn on the basis of correlational evidence and a lack of mediation evidence are limited, these findings broadly capture the theories of stress reduction and attention restoration. The identified outcomes of nature exposure reported in the current review were in line with relevant theories. Decreases in stress and improvements in positive and negative mood states are in line with SRT. Improvements in positive and negative moods states, positive affect and anxiety are also in line with an emotional regulation theory of nature’s benefits on psychological outcomes. Outcomes of improved vitality and perceived restoration are in line with ART. However, as the theories of SRT, ART and emotional regulation highlight multiple possible pathways of effect, research studies operationalise the effects of nature exposure through varying indictors and findings indicate that the theories are not mutually exclusive [114]. More theoretical development is therefore needed, more specifically through the appropriate use of theory-guided statistical pathway modelling, to explore interdependencies of pathways often treated as individually within the literature [112] and to learn more about the key pathways of effect. Furthermore, findings from Scarf et al. [97] suggest that social constructs might interact with or be independent of the mechanisms supposed in the above theories. Firm conclusions on key mechanisms, however, cannot be made, given that the studies in the current review are methodologically limited and there is a paucity of mediation analyses.

### 4.1. Strengths

The current review, to our knowledge, is the first systematic review that aimed to review any potential mechanisms of nature in promoting psychological change in the adolescent literature, specifically in experimental and quasi-experimental studies with a control condition only. This builds on previous reviews where the evidence base has been dominated by cross-sectional studies [23]. Such designs, that include a nature-based manipulation, enable a better estimation of nature’s effect on psychological change, accounting for more confounding variables than in cross-sectional research. Furthermore, the inclusion of studies with control conditions strengthens the findings drawn and conclusions made. By excluding studies with integrated mental health or wellbeing components, the review aimed to distinguish any mechanisms of nature exposure on psychological change independent of therapeutic input. 

### 4.2. Limitations 

The current review focused only on peer-reviewed articles, meaning grey literature and unpublished studies were not included, which may limit the generalisability of the findings. The quality of the literature was low overall. Only 11 out of 27 papers were rated as being of moderate methodological quality and used within the synthesis of results. The remaining included papers were classified as low quality, so conclusions from these papers were not drawn. In considering the categories of the quality assessment tool that supported weak quality ratings blinding should be carefully considered as this category was rated as weak in the majority of the 16 weakly rated studies. Full blinding in nature-based research is not possible due to being unable to blind participants to environmental conditions. Future systematic reviews in this area may wish to consider an alternative quality assessment tool or assign weights to the categories included so that ratings of blinding do not lead to the exclusion of important findings. 

Some important methodological limitations and risk of bias also existed in the moderately rated studies, meaning that the findings should be interpreted with caution. Eight of the moderately rated papers used undergraduate samples only, which captures a smaller range of adolescents (age 18–24) and therefore limits the extent to which findings can be applied to adolescents of a younger age range. Two of these studies [90] used a male undergraduate sample only and the majority of the other undergraduate samples were imbalanced with regard to gender. This is of particular importance as previous reviews have noted the relationship between nature and outcomes to vary by demographic factors such as gender [23,115]. Given that there may be gender differences in outcomes, conclusions drawn across genders may be limited. 

The selection of study criteria may have influenced the limited findings of mechanisms of change. Whilst the selection of controlled, experimental studies was applied as a stringent identification of mediators given problems with non-experimental data in undermining assumptions of the statistical mediation model [116], as previously stated, the nature evidence base in an adolescent population is dominated by cross-sectional data. The selection criteria was therefore potentially too demanding, meaning that papers reporting mediation analyses were not included. 

### 4.3. Implications and Future Research

Understanding mediating variables is imperative in investigating why or how experimental manipulations and interventions lead to change, highlighting potential mechanisms of change [109]. The identification of mediators is an important first step of intervention development, leading to subsequent randomised control trials enhanced in components associated with the mediator to establish key targetable mechanisms [117]. Given the paucity of controlled experimental studies with the inclusion of mediation analyses in the current review, future research should focus on highlighting the potential mediators of nature exposure on wellbeing in adolescents. Whilst the identified outcomes of psychological change in the current review may be plausible mechanisms, the evidence is of low methodological quality. There is a need to test and clarify these outcomes as potential mediators in robustly controlled, experimental studies with longitudinal designs. Furthermore, given that nature has benefits via a seemingly complex web of mechanisms [2], there may be other mechanisms outside what was indicated in the current study. 

Whilst outside the scope of this review, the methodological limitations of the included studies highlight the need to explore whether mechanisms of change differ within subpopulations of adolescence. Identification of these moderating variables is crucial in understanding how nature exposure might impact on sub-groups differently [118]. Given the diverse outcomes highlighted in the current review, the complexity of human-nature relations and potential differential effects according to moderating variables, it is entirely possible that multiple pathways exist. Ongoing research should aim to include advance multi-variate modelling to consider such complexity, specifically ‘what works for whom’. 

Given the heightened potential for nature exposure as a modifiable lifestyle behaviour [13] nature may be a targetable factor in clinical practice for the prevention of mental health problems and requires further exploration [16]. Outcomes of mood, affect and stress in the current review are of particular interest as they are also implicated in the development and maintenance of mental health difficulties such as depression in adolescence. Paying attention to these as potential mechanisms in future research would allow for the development of clinically relevant nature-based interventions, including ‘active ingredients’ in an adolescent population. This would enable the design of interventions with greater clinical effectiveness. 

## 5. Conclusions

This review systematically searched for evidence from controlled experimental research of key mechanisms of psychological change following nature exposure in an adolescent population. With a particular lack of statistical mediation models within the evidence, it is not possible to make any firm conclusions about the key mechanisms in this population. Given the current quality of the evidence, whilst outcomes of psychological change following nature exposure have been outlined, there is a need for further research to design robust and longitudinal, experimental studies to test and evaluate for plausible mechanisms. Consideration of mediation and moderation pathways will be important for specifying what works best and for whom in an adolescent population, in order to inform the development of future nature-based interventions to support mental health and wellbeing.

## Figures and Tables

**Figure 1 ijerph-19-12649-f001:**
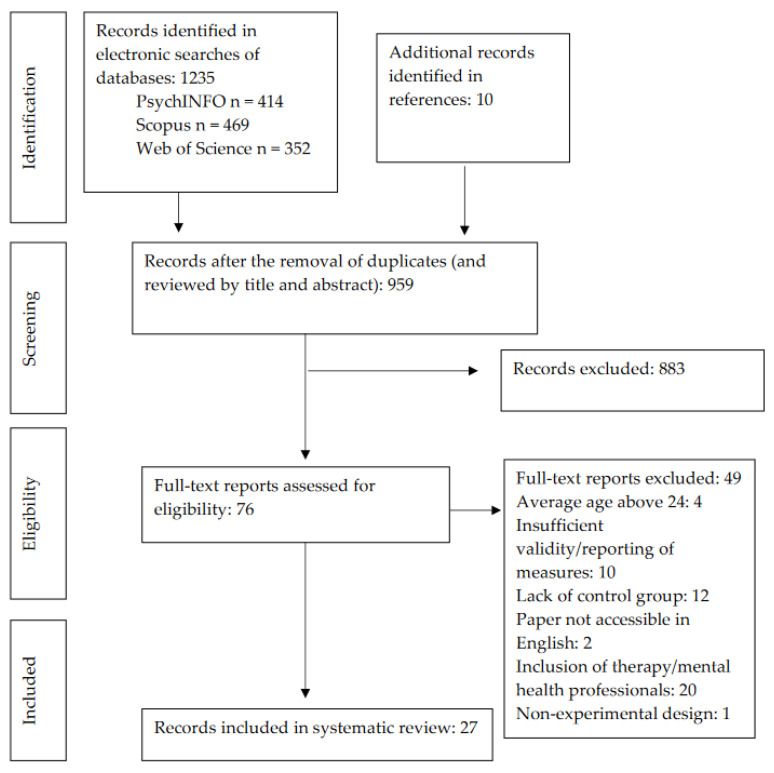
Study Selection Process Based on PRISMA Guidelines.

**Table 1 ijerph-19-12649-t001:** Search Terms with Boolean Operators.

Key Word		Search Terms
Nature		“contact with nature” OR “nature exposure” OR “exposure to nature” OR “natur * experience *” OR “access to nature” OR “green space *” OR greenspace OR greenery OR greenness OR forests OR “forest school” OR “shinrin yoku” OR “shinrin-yoku” OR “forest bathing” OR “forest environment” OR wilderness OR “blue space” OR park OR parks OR woodlands OR countryside OR “urban forest” OR “outdoor adventure interventions” OR “adventure therapy” OR gardening OR “natur * environment *” OR “outdoor adventure education” OR “adventure education” OR “adventure program” OR outdoors OR “green exercise” OR “nature therap *” OR “green play” OR “nature therap *” OR ecotherapy * OR “school landscape”
Mental Health	AND	“mental health” OR anxiety OR depress * OR mood OR well-being OR wellbeing OR well-being OR “strengths and difficulties”
Adolescence	AND	Adolescen * OR teen * OR “young people” OR “young adult *” OR youth *

**Table 2 ijerph-19-12649-t002:** Inclusion and Exclusion Criteria Based on the PICOs Question.

PICOS	Inclusion Criteria	Exclusion Criteria
*Population*	Samples with an average age of 24 and under	
*Intervention*	Exposure to all types of nature	A therapeutic intervention or intervention delivered by trained mental health professionals
*Comparison*	All comparison groups	A lack of control group
*Outcomes*	Changes in mental health and psychological status	
*Study Design*	Experimental or quasi-experimental	

**Table 3 ijerph-19-12649-t003:** Study Characteristics.

Paper number (#)	Authors (Year)	Country	N	Age in years Range (Mean ± SD)	Gender of Sample	Study Design	Nature Exposure (Experimental)	Comparison	Quality Rating
1	Bielinis et al. (2019) [74]	Poland	32	(20.97 ± 0.65)	Female	Experimental	20 min walk, 15 min exposure to snow covered forest	20 min walk to uban environment	Mod
2	Greenwood & Gatersleben (2016) [75]	UK	120	16–18 (-)	Mixed	Experimental	20 min grassed quandrangle: alone, with friend, with phone	Indoor for 20 min	Weak
3	Hassan et al. (2018) [76]	China	60	19–24 (19.60 ± 1.42)	Mixed	Cross-over experimental	5 min rest, 15 min guided walk in bamboo forest	5 min rest, 15 min guided walk in city area	Weak
4	Bielinis et al. (2021) [77]	Poland	22	(22.50 ± 4.67)	Mixed	Cross-over experimental	15 min viewing in snow covered forest	15 min viewing building landscape	Weak
5	Bielinis, Omelan et al. (2018) [78]	Poland	54	(21.35 ± 1.39)	Mixed	Experimental	15 min viewing in forest in winter vs. spring	Urban setting in winter vs. in spring	Mod
6	Bielinis, Takayama et al. (2018) [79]	Poland	62	(21.45 ± 0.18)	Mixed	Experimental	15 min walk, 15 min viewing in forest	15 min in Urban environment	Mod
7	Hartig et al. (2003) [80]	USA	112	(20.8 ± 3.7)	Mixed	Experimental	Task vs. no task in vegetation and wildlife preserve	Task vs. no task in urban environment	Mod
**(#)**	**Authors (Year)**	**Country**	**N**	**Age in years** **Range** **(Mean ± SD)**	**Gender of Sample**	**Study Design**	**Nature Exposure (Experimental)**	**Comparison**	**Quality** **Rating**
8	Lee et al. (2011) [81]	Japan	12	(21.20 ± 0.9)	Male	Cross-over experimental	15 min forest viewing	15 min urban viewing	Weak
9	Lee et al. (2014) [82]	Japan	48	(21.10 ± 1.2)	Male	Cross-over experimental	Forest walking and self-paced walking in forest	Urban walking and self-paced walking in urban environment	Weak
10	Mao et al. (2012) [83]	China	20	(20.79 ± 0.54)	Male	Experimental	2 × 1.5 h walks in forest area	2 × 1.5 h walks in city area	Weak
11	Park et al. (2011) [84]	Japan	168	(20.40 ± 4.1)	Male	Cross-over experimental	15 min viewing, 15 min walk in forest (14 forests)	15 min viewing, 15 min walk in urban area (14 areas)	Weak
12	Park et al. (2010) [85]	Japan	280	(21.70 ± 1.5)	Male	Cross-over experimental	15 min viewing, 15 min walk in forest (24 forests)	15 min viewing, 15 min walk in urban area (24 areas)	Weak
13	Song et al. (2014) [86]	Japan	17	(21.20 ± 1.7)	Male	Cross-over experimental	15 min walk in urban park	15 min walk in city area	Weak
14	Song et al. (2015) [87]	Japan	23	(22.3 ± 1.2)	Male	Cross-over experimental	15 min walk in urban park	15 min walk in city area	Weak
15	Song et al. (2013) [88]	Japan	13	(22.50 ± 3.1)	Male	Cross-over experimental	15 min walk in urban park	15 min walk in city area	Weak
16	Takayama et al. (2011) [89]	Japan	45	(21.21 ± 1.25)	Male	Cross-over experimental	15 min walk (morning), 15 min viewing (afternoon) in forest	15 min walk (morning), 15 min viewing (afternoon) in urban area	Mod
**(#)**	**Authors (Year)**	**Country**	**N**	**Age in years** **Range** **(Mean ± SD)**	**Gender of Sample**	**Study Design**	**Nature Exposure (Experimental)**	**Comparison**	**Quality** **Rating**
17	Tsunetsugu et al. (2013) [90]	Japan	48	(21.10 ± 1.1)	Male	Cross-over experimental	15 min viewing in forest	15 min viewing urban site	Mod
18	McAnally et al. (2018) [91]	New Zealand	106	(14.43)	Male	Quasi-experimental	2 terms of outdoor education	Education as usual	Weak
19	Fuegen & Breitenbecher (2018) [92]	USA	108	17–75 (21.59 ± 7.69)	Mixed	Experimental	Outdoor exercise and outdoor rest (university grounds)	Indoor exercise vs. indoor rest (simulated)	Mod
20	Shin & Oh (1996) [93]	Korea	32	18–32 (23.13)	Mixed	Quasi-experimental	5-day forest program	Wait list control	Weak
21	Wang et al. (2016) [94]	China	140	18–24 (22.38 ± 2.56)	Mixed	Experimental	Exposure to video tapes of urban parks during stress recovery	Video tapes of urban roadways	Mod
22	Wood et al. (2013) [95]	UK	25	(13.10 + 0.3)	Mixed	Cross-over experimental	Exercise whilst viewing outdoor natural scene	Exercise viewing built up environment	Mod
23	Kelz et al., (2015) [96]	Austria	195	10–18 (14.40)	Mixed	Quasi-experimental	Access to green schoolyard	Control schools	Mod
24	Scarf et al. (2017) [97]	New Zealand	180	(16.54)	Mixed	Quasi-experimental	10-day developmental voyage on a ship	Education as usual	Weak
25	Scarf et al. (2016) [98]	New Zealand	180	15–19 (16.56)	Mixed	Quasi-experimental	10-day developmental voyage on a ship	Education as usual (but two different groups at T1 and T4)	Mod
**(#)**	**Authors (Year)**	**Country**	**N**	**Age in years** **Range** **(Mean ± SD)**	**Gender of Sample**	**Study Design**	**Nature Exposure (Experimental)**	**Comparison**	**Quality** **Rating**
26	Hayhurst et al. (2015) [99]	New Zealand	(1): 120 (2): 146	(1): (17.98) (2):(16.47)	Mixed	Quasi-experimental	10-day developmental voyage on a ship	No voyage	Weak
27	Hunter et al. (2013) [100]	New Zealand	(1): 62 (2): 396	(1): (16.46)(2): (16.62)	Mixed	Quasi-experimental	10-day developmental voyage on a ship	Education as usual	Weak

Note. Mod = Moderate. Quality ratings are according to EPHPP quality appraisal assessment.

**Table 4 ijerph-19-12649-t004:** Significant outcomes of psychological change following exposure to nature identified in each of the studies.

Paper number (#)	Authors	Outcomes	Significant Results
1	Bielinis et al. (2019). [74]	Tension/Anxiety (POMS)	Decreased after exposure to snow-covered forest environment compared to urban forest environment (*F* = 18.06, *p* = 0.000, *η*^2^ = 0.37).
		Depression/Dejection (POMS)	Decreased after exposure to snow-covered forest environment compared to urban forest environment (*F* = 7.315, *p* = 0.011, *η^2^* = 0.20)
		Anger/Hostility (POMS)	Decreased after exposure to snow-covered forest environment compared to urban forest environment (*F* = 16.198, *p* = 0.000, *η^2^* = 0.35)
		Confusion (POMS)	Decreased after exposure to snow-covered forest environment compared to urban forest environment (*F* = 9.172, *p* = 0.005, *η^2^* = 0.23)
		Negative affect (PANAS)	Increased after exposure to the Urban environment compared to forest environment (*F* = 4.999, *p* = 0.033, *η^2^* = 0.14)
		Restoration (Restorative Outcome Scale)	Increased after exposure to the forest environment compared to urban environment (*F* = 8.885, *p* = 0.006, *η^2^* = 0.23)
		Vitality (Subjective Vitality Scale)	Increased after exposure to the forest environment compared to urban environment (*F* = 4.527, *p* = 0.042, *η^2^* = 0.13)
5	Bielinis et al. (2018a) [78]	Tension/Anxiety (POMS)	Significant effect of experimental interventions (*F =* 7.47, *p* < 0.001). Lowest values observed in forest in the winter.
		Depression/Dejection (POMS)	Significant effect of experimental interventions (*F =* 5.49, *p* < 0.001). Lowest values observed in forest in the winter.
		Anger/Hostility (POMS)	Significant effect of experimental interventions (*F =* 4.25, *p* < 0.001) Lowest values observed in forest in the winter.
		Fatigue (POMS)	Significant effect of experimental interventions (*F =* 4.79, *p* < 0.001). Lowest values observed in forest winter and forest spring interventions.
		Confusion (POMS)	Significant effect of experimental interventions (*F =* 5.18, *p* < 0.001). Lowest values observed in forest in the winter.
**(#)**	**Authors**	**Outcomes**	**Significant Results**
	Bielinis et al. (2018a) [78] cont.	Vigour (POMS)	Significant effect of experimental interventions (*F =* 4.96, *p* < 0.001). Values were significantly higher in the forest environment during the winter than in the room or city conditions.
		Positive Affect (PANAS)	Significant effect of experimental interventions (*F* = 4.34, *p* < 0.001)Significantly higher in the forest environment during both seasons.
		Restoration (Restorative Outcome Scale)	Significant effect of experimental interventions (*F =* 6.31, *p* < 0.001). Highest values were observed in the forest during the winter and during the spring, but during the winter they were significantly higher than during the spring
		Vitality (Subjective Vitality Scale)	Significant effect of experimental interventions (*F =* 5.37, *p* < 0.001). Values in the forest during the winter were significantly higher than during the spring, or in the room, or the city conditions.
6	Bielinis et al. (2018b) [79]	Tension/Anxiety (POMS)	Significantly lower scores observed in the forest vs. urban environment (*F* = 45.49, *p* = 0.000).
		Depression/Dejection (POMS)	Significantly lower scores observed in the forest vs. urban environment (*F =* 22.09, *p* = 0.000).
		Anger/Hostility (POMS)	Significantly lower scores observed in the forest vs. urban environment (*F =* 25.35, *p* = 0.000).
		Fatigue (POMS)	Significantly lower scores observed in the forest vs. urban environment (*F =* 28.1, *p* = 0.000).
		Confusion (POMS)	Significantly lower scores observed in the forest vs. urban environment (*F =* 20.3, *p* = 0.000).
		Vigour (POMS)	Significantly higher scores observed in the forest vs. urban environment (*F =* 28.35, *p* = 0.000).
		Positive Affect (PANAS)	Significantly higher scores observed in the forest vs. urban environment (*F =* 17.01, *p* = 0.000).
		Negative Affect (PANAS)	Significantly higher scores observed in the urban vs. forest environment (*F =* 15.18, *p* = 0.000).
		Restoration (Restorative Outcome Scale)	Significantly higher scores observed in the forest vs. urban environment (*F =* 35.27, *p* = 0.000).
		Vitality (Subjective Vitality Scale)	Significantly higher scores observed in the forest vs. urban environment (*F =* 27.68, *p* = 0.000).
7	Hartig et al. (2003) [80]	Diastolic Blood Pressure	Subjects with tree views showed significantly steeper DBP declines than the subjects in a viewless room (*F*(2, 180) = 4.74, *p* = 0.01)
		Systolic Blood Pressure	A significant environment X time interaction in the analysis of the readings at 20, 30, 40, and 50 min (*F*(3, 249) *=* 2.94, *p* = 0.04).
		Emotion (Zucker’s Inventory of Personal Reactions) emotion	Subjects walking in the nature reserve experienced more positive emotion than those walking in the urban environment (*F*(1, 49) = 7.40, *p* = 0.01).
		Positive Affect (PANAS)	Positive affect increased at the nature reserve and decreased in the urban environment (*F*(1, 100) *=* 56.83, *p* < 0.001).
**(#)**	**Authors**	**Outcomes**	**Significant Results**
	Hartig et al. (2003) [80] cont.	Attention (Necker Cube Pattern Control Task)	Performance improved in the natural environment but suffered in the urban environment, regardless of antecedent condition. (*F*(1, 98) *=* 13.15, *p* < 0.001).
16	Takayama et al. (2011) [89]	Tension/Anxiety (POMS)	Significantly lower in the forest environment than in the urban areas (p < 0.000).
		Fatigue (POMS)	Significantly lower in the forest environment than in the urban areas (*p* = 0.000).
		Confusion (POMS)	Significantly lower in the forest environment than in the urban areas (*p* = 0.000).
		Vigour (POMS)	Significantly higher in the forest environment after the viewing session (*p = 0.000*).
		Positive Affect (PANAS)	Significantly higher in the forest environment than in the urban areas (*p* = 0.001).
		Negative Affect (PANAS)	Significantly lower in the forest environment than in the urban areas (*p* < 0.000).
		Restoration (Restorative Outcome Scale)	Significantly higher in the forest environment than in the urban areas (*p* < 0.000).
		Vitality (Subjective Vitality Scale)	Significantly higher in the forest environment after walking (*p* < 0.000) and after the combined effect of walking and viewing (*p* < 0.000).
17	Tsunetsugu et al. (2013) [90]	Diastolic Blood Pressure	Significantly lower in the forested areas than in the urban areas *(**p* = 0.034, η^2^_p_ = 0.10).
		Heart Rate Variability (High Frequency)	Continuously significantly higher in the forested areas (*p* < 0.01, *d =* 0.31–0.70).
		Heart Rate	Significantly lower in the forested area during every minute of viewing than in the urban areas *(**p* < 0.01, *d* = 0.49–0.71).
		Tension/Anxiety (POMS)	Significantly increased when viewing the scenery in urban areas (*p* = 0.00, η^2^_p_ = 0.22).
		Fatigue (POMS)	Significantly increased when viewing the scenery in urban areas (*p* = 0.00, η^2^_p_ = 0.35).
		Confusion (POMS)	Significantly increased when viewing the scenery in urban areas (*p* = 0.01, η^2^_p_ = 0.35).
		Vigour (POMS)	Significantly decreased when viewing the scenery in urban areas (*p* = 0.00, η^2^_p_ = 0.26).
19	Fuegen & Breitenbecher (2018) [92]	Positive Affect (PANAS)	Participants whose sessions took place outdoors experienced a slight increase in positive affect (*F*(1, 171) = 22.54, *p* < 0.001, η^2^_p_= 0.12).
		Energy (AD-ACL)	Participants whose sessions took place outdoors experienced an increase in energy (*F*(1, 174) *=* 18.99, *p* < 0.001, η^2^_p_ = 0.10).
		Tiredness (AD-ACL)	Participants whose sessions took place outside experienced a decrease in tiredness *(**F*(1, 175) = 12.10, *p* = 0.001, η^2^_p_ = 0.07).
21	Wang et al. (2016) [94]	Skin Conductance	Significant differences scenes (χ2(6) *=* 22.379, *p* = 0.001, η^2^= 0.16). Compared with viewing the Urban Roadway, subjects’ mean SCR values were significantly reduced by viewing Lawn w/people (*p* = 0.005), Lawn w/o people (*p* = 0.006), Small Lake (*p* = 0.022) and Walkway (*p* = 0.022).
**(#)**	**Authors**	**Outcomes**	**Significant Results**
	Wang et al. (2016) [94] cont	Electrocardiogram (R-R intervals)	Significant effect of viewing different sites on length of R–R intervals (*F* (6, 126) = 2.499, *p* = 0.026, η^2^= 0.10).R–R intervals increased significantly more Lake (0.116 ± 0.06 s, *p =* 0.047), compared to viewing the Urban after viewing the Walkway (0.125 ± 0.06 s, *p* = 0.010) and Small Roadway (0.063 ± 0.06 s).
		Attention (Digit Span Backwards)	Participants’ attentional levels improved significantly after watching Lawn with people (*p* < 0.001, d = 1.09), Lawn without people (*p* = 0.001, *d* = 0.97 ), Plaza without people: (*p* < 0.001, *d =* 1.17), Small Lake (*p* = 0.007, *d* = 0.58),Walkway (*p* = 0.001, *d* = 0.73)
		State Anxiety (State-trait Anxiety Inventory)	All six urban park scenes had a significant positive effect on state-anxiety relief, compared with the Urban Roadway scene (F(6, *F*(6, 133) = 11.59, *p* < 0.001, *η^2^* = 0.31)
		Restoration (Perceived Restorativeness)	Significant differences among the seven scenes (*F*(6, 133) = 25.68, *p* < 0.001, *η^2^*= 0.54). All six urban park scenes were perceived as more restorative than the Urban Roadway scene (*p* < 0.001)
23	Kelz et al. (2015) [95]	Diastolic Blood Pressure	Significantly lower for the experimental school’s pupils at the second time of measurement compared with the mean of both times of the control school’s measurements and the experimental school’s first time of measurement (*F*(1, 184.3) = 15.46, *p* = 0.001, *d* = 0.41).
		Systolic Blood Pressure	Significantly lower for the experimental school’s pupils at the second time of measurement compared with the mean of both times of the control school’s measurements and the experimental school’s first time of measurement (*F*(1, 175.4) = 5.14, *p* = 0.025, *d* = 0.23).
		Wellbeing (Intro-psychic Balance)	Pupils from the school had significantly higher scores after the installation of the schoolyard compared with the mean of both times of measurement at the control school and the first time of measurement at the experimental school. (*F*(1, 175.4) = 5.14, *p* = 0.025, *d =* 0.23).
		Wellbeing (Recovery-stress Questionnaire)	Pupils from the school had significantly higher scores after the installation of the schoolyard compared with the mean of both times of measurement at the control school and the first time of measurement at the experimental school. (*F*(1, 172.3) = 3.78, *p* = 0.053, *d =* 0.18).
		Restoration (Perceived Restorativeness)	Perceived restoration increased pre- to post-renovation for measures of compatibility only (*t*(62) = 3.86, *p* = 0.001, *d* = 0.48).
**(#)**	**Authors**	**Outcomes**	**Significant Results**
25	Scarf et al. (2016) [98]	Resilience (Resilience Scale)	Significantly improved overtime (*F*(1, 59) = 102.54, *p* < 0.001, *η^2^* = 0.63).

Note. *η^2^* = partial eta squared.

## Data Availability

Please contact the authors directly to request datasets for the study.

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
