# Peer review of "A Systematic Review of Mechanisms Underpinning Psychological Change Following Nature Exposure in an Adolescent Population"

_ijerph, 2022, doi:10.3390/ijerph191912649_

Round 1

Reviewer 1 Report

Thank you for the opportunity to review your manuscript, which reports a very sound systematic review following the PRISMA Guidelines. The focus of the review on mechanisms of psychological change following exposure to nature within an adolescent population is in important step towards elucidating specific foundations for change induced by exposure to nature at this developmental phase of life.

Given your focus on the period of adolescence, it seemed odd to me that there was no direct reference to the age-related changes in nature connectedness documented in the literature. For example, Richardson et al. (2019) reported age-related changes in nature connectedness, with a drop in levels of nature connectedness in children between the ages of 10-15 years reducing to lowest levels amongst those in the 13-18-year age range and an increase back to childhood levels occurring throughout early adulthood until peaking again around the age of 31 years. A Slovakian study reported a lack of positive attitudes towards plants in children between the ages of 10-15 years, although those age-related changes were inconsistent across four dimensions of attitudes towards plants (Fančovičová & Prokop, 2010). Kaplan and Kaplan (2002) also reported a ‘time out’ period during adolescence when preferences for nature settings waned in favour of places where teens could exercise their social orientation and test their self-determination.

Otherwise, I enjoyed reading the review and particularly the take-aways for future research, which I believe will be of use to those of us working in this field.

I have inserted highlights within the manuscript either to indicate where attention needs to be directed to wording, or to flag some specific comments or queries for you to take into consideration.

Lines 132-135. This sentence is overly long and complex; would be better broken down into two or more sentences for clarity.

Lines 377-378. Seems you’re saying that outcomes were identified and assigned.

Line 402. Indicate author name/s.

References. Multiple entries require correction.

Author Response

Dear Reviewer,

Thank you for kindly taking the time to review our submission and for taking the time to highlight the required changes within the manuscript.

Your comment on the omission of research indicating changes in nature connectedness was very insightful and perhaps an oversight on our behalf. It will be something that we hold in mind for any future work and discussion in this area. 

We have amended the manuscript with track changes (document attached) so that you can clearly see where we have amended based on your feedback. Please also find below a comment associated with the individual amendments on specified lines. Where there was a specific reviewer comment in the feedback (in additional to being highlighted in the manuscript), this has been written in bold text.

Line 42 – We have added a full stop at the end of the sentence.

Line 82 – Cofounders has been amended to read confounders

Line 84 – Resillience has been amended to read resilience

Reviewer comment: Lines 132-135. This sentence is overly long and complex; would be better broken down into two or more sentences for clarity:

Line 133-137 have been amended into two separate sentences so that it now reads:

“Firstly, perceived restorativeness [47–49] has been suggested, which is a measure of the restorative components of natural environments as indicated by ART, but is not necessarily predictive of or necessary for attention restoration [50]. Secondly, nature connectedness as a psycho-evolutionary need to affiliate with the natural environment has also been suggested as an additional construct [51].”

Line 140 – We have added in spaces and a comma so that it now reads: mindfulness [43],

Line 213 – An apostrophe has been added to authors recommendations so that it now reads: authors’ recommendations

Line 231 – We have moved the full stop from line 232 to end the sentence on 231 correctly

Line 261 – Studies has been changed to read: each study

Line 262 – Week has been amended to read: weak

Line 287 – Authors has been added in, to aid readability: “however, the authors used…”

Lines 377-378. Seems you’re saying that outcomes were identified and assigned.

Line 391 – We have re-written the sentence to aid readability: “Outcomes associated with psychological change were then identified and subsequently assigned to 11 categories.”

Line 405 – We have removed the additional spacing so that the sentence is formatted on one line

Line 415 – We have inserted the name of the authors to improve the readability: “on stress recovery, Corazon et al. [31] also found”

Line 431 – We have removed the word “Han” to read: “potential mediation effects [109].”

Line 487 – We have changed the semi-colon to a comma: “evidence are limited,”

Line 492 – In fitting has been amended to: “in line”

Line 499 – “Dzhambov” has been removed

Line 501 – We have added in the word with so that it reads: “might interact with or be independent of”

Line 572-673 – Behavioural has been amended to behaviour

Table Formatting – the table formatting has been amended throughout to ensure that the tables are headed when they carry on over onto the next page

References – Thank you for going through the references so thoroughly. We have re-checked every reference and amended where required

We would like to thank you once again for the helpful comments and suggestions on our manuscript. After carefully considering all points raised, we believe the amended manuscript is now much improved. We look forwarded to hearing from you again.

Yours Sincerely,

Megan Rowley

Reviewer 2 Report

This is an important research project to define the effects of nature on the emational and behavior of adolescents especially with the volume of comment on the mental health issues of this group.  The research aims of qualitative research into the changes that can occur when the natural environment is part of a series of methods to assist in maintaining the mental health of adolescents.  The methodology in collecting the relevant published data is excellent and approprite for this study while the analysis invoking the theories Stress Reduction Theory, Attention Restoration Theory, and Affect Regulation offer an excellent analysis method for the collected data.  The tabulated data and additional information provide clarity of research methods and add validity to the research.  The analysis and conclusions are plausable and well-constructed from the collected data.

One minor criticism: Line 225.  It is better practice not to commence a sentence with a numeral.

This is a worthy research project that appears to have required major input from the authors for which they are commended.

Author Response

Dear Reviewer

Thank you for kindly taking the time to review our submission and for your thoughtful comments. 

In line with your feedback I have now amended line 225 from "76 papers were included for full-text review and 37 met eligibility criteria" to "A total of 76 papers..." 

After carefully considering your points raised, we believe the amended manuscript is now improved. We look forwarded to hearing from you again.